# Digital Health Services through Patient Empowerment: Classification, Current State and Preliminary Impact Assessment by Health Pod Systems

Giuseppe Andreoni [1,2,*], Enrico Gianluca Caiani [3,4] and Nicola Castaldini [5]

1   Department of Design, Politecnico di Milano, 20158 Milan, Italy
2   Istituto di Bioimmagini e Fisiologia Molecolare, Consiglio Nazionale delle Ricerche, 20090 Segrate, Italy
3   Department of di Electronics, Information and Bioengineering, Politecnico di Milano, 20133 Milan, Italy;
    enrico.caiani@polimi.it
4   Istituto di Elettronica e di Ingegneria dell'Informazione e delle Telecomunicazioni, Consiglio Nazionale delle
    Ricerche, 20133 Milan, Italy
5   Maria Cecilia Hospital, GVM Care & Research, 48033 Cotignola, Italy; ncastaldini@gvmnet.it
*   Correspondence: giuseppe.andreoni@polimi.it

**Featured Application:** The innovative systems called health pods are exploiting digital health concepts by delivering preventive medicine services in a "distributed hospital" setting.

**Abstract:** Health pods are new systems such as small spaces equipped with medical devices where users can measure several biomedical parameters related to their health status and receive other medical services. Their impact on health over a life course could be relevant in defining healthy aging strategies and/or management of chronic diseases and the early detection of possible symptoms related to some common pathologies. The generated data have not only a personal value but even at a community/society level. Health pods also support educational and empowerment actions to enable the 5P medicine approach, and specifically prevention, health promotion, and public health policymaking. This paper aims at defining their taxonomy, conducting a market and typologies survey, and discussing their potential impact in preventive medicine, presenting data of a pilot test carried out placing two health pods in a superstore environment to validate the demand and the participation of people in a prevention campaign. A 57-day period was observed at two sites: the number of free accesses and administered tests was impressive for size and completeness. The test revealed a good picture of the general health status of the population, with satisfying AGE values in the cardiovascular check and stress index through an HRV analysis. The body composition test revealed a small number of overweight subjects, more in males than in females. This pilot confirmed the huge demand for personalized services for improving well-being, health status, and quality of life and the relevance of these solutions for their individual and societal impact in preventive medicine.

**Keywords:** health pod; lifestyle medicine; patient empowerment; health promotion; prevention; opportunistic screenings

## 1. Introduction

### 1.1. The Reference Scenario in Healthcare during the Current Decade

The worldwide situation in healthcare is profoundly marked by two great phenomena:
- the "silverization" of society
- the continuous growth of healthcare expenditure that is reaching serious and warning levels.

In 2020, the pandemic emergency caused by the SARS-CoV2-2019 virus added to these stable trends. The demographical analysis is clearly showing the expectation for societal age distribution in the coming decades, highlighting an emergency for the cost of managing

chronic disease related to aging. Only in Europe, the rate of people above 65 years old will increase from 20% of the total population in 2020 to a projection of 30% in 2050 [1]. Similarly, the number of people potentially in need of long-term care is expected to increase from 19.5 M in 2016 to 23.6 M in 2030 and 30.5 M in 2050 (EU-27). An aging society implies a larger population of chronically impaired people that need cures and care, thus impacting negatively on healthcare costs (Eurostat [2]) and imposing a considerable economic burden on health services. Indeed, chronic diseases are a major public health problem worldwide. By 2030, the proportion of total global deaths due to chronic diseases is expected to increase to 70%, and the global burden of disease to 56%, with the greatest increase anticipated in the African and Eastern Mediterranean regions [3]. Non-communicable diseases (NCDs) are the major causes of death worldwide and underlie almost two-thirds of all global deaths [4,5]. Although all countries face high numbers of these diseases, low-income and middle-income countries and the poorest and most vulnerable populations within them are affected the most. There is a global imperative to create and implement effective prevention strategies because the future costs of healthcare are likely to be unaffordable [6].

*1.2. Lifestyle and the 5P Medicine Approach*

From recent scientific findings, 84% of NCDs have a direct correlation with lifestyle. Lifestyle diseases share risk factors similar to prolonged exposure to four modifiable lifestyle behaviors—smoking, unhealthy diet, stress, and physical inactivity—and result in the development of chronic diseases, specifically heart disease, stroke, diabetes, obesity, metabolic syndrome, chronic obstructive pulmonary disease, and some types of cancer [7–9]. Contrary to popular belief, genetics affect their development by 30–35%, against 60–65% attributed to lifestyle [10,11]. Their genesis, in fact, is determined for the most part by a cluster of risk factors common to all, which, moreover, act toward each other as an amplifier of noxious effect. These risk factors are high blood pressure, dyslipidemia, sedentary lifestyle, overweight, tobacco use, and alcohol abuse. The WHO estimates that each of these factors determines a loss of life expectancy ranging from 12 years for smoking to 3.3 years for a sedentary lifestyle, without considering the overlapping effects between several factors [12]. Moreover, most of these conditions are asymptomatic for many years before manifesting themselves in a clinically overt way. Fortunately, most of the risk factors mentioned above are reversible and, if corrected in time, so are the diseases they trigger. For this reason, in the "healthy" population, it is essential to early identify their presence and correct them in order to actively prevent the appearance of the "deadly quartet" described above.

In this frame, together with the already established preventive medicine, today a new clinical branch of "lifestyle medicine" is starting, in which the definition, promotion, and personalization of coaching interventions are the most promising strategies. To reach this proactive healthcare promotion goal, the most important action is considered to be patient empowerment, defined as the process by which people can gain better control over decisions and actions related to their own health [13]. This is the vision for healthcare adopted by the European Union in its key strategies for research and related exploitation, in which digital transformation and trends toward predictive, preventive, personalized, and participatory (the 4P approach) healthcare are set as targets for 2030 [14]. More recently, a fifth P was added to include the psycho-cognitive [15,16] or pluri-expert aspects, thus resulting in the so-called P5 medicine. Both 4P and 5P definitions recognize the multidimensional approach needed for a successful outcome.

Predictive medicine is built on two pillars: (1) genetics and genomics and (2) lifestyle. The preventive lifestyle interventions must be transformed from actions planned on clusters of the population to tailored interventions for the single individual to achieve the best effectiveness, efficacy, quality, and acceptance, also using psycho-cognitive behavioral change models. Personalization to the individual user requires his/her participation, i.e., an active role of patients in the co-creation of systems, strategies, therapies, and interventions to be really user-centered for their best results. Furthermore, participatory medicine also

means enabling the subject to have an active role in the management of her/his own health, starting from awareness and knowledge of their current health status and personal lifestyle. Moreover, since 2020, the COVID-19 pandemic has dramatically demonstrated the urgent need for personal responsibility in adopting safe behaviors, for digital services to assure continuity of care, surveillance/monitoring or remote assisted intervention, and, above all, tools for population screening and personal protection tools/services.

With these premises, to intervene effectively in the prevention and early treatment of these pathologies, it is essential to shift attention outside the healthcare facilities to the general population, to identify those at risk or already affected, and to be able to treat them as soon as possible, thereby reducing the actual disease burden.

### 1.3. Innovative Technologies for Prevention

Currently, the state of available technologies for patient empowerment includes wearable technologies, un-obtrusive monitoring (i.e., environmental sensing), and several devices dedicated to specific measurements of physiological parameters. If wearable technologies offer continuous monitoring of some basic functional parameters, a deep insight into health parameters is often needed. The recent concept of the health care pod or kiosk has been proposed by some researchers to tackle this aspect [17–20]. However, a commonly agreed definition for these solutions is still lacking, according to the different delivered services and even to their different physical configurations.

Therefore, the aim of this paper is threefold in relation to three different aspects.

From a methodological point of view, there is the need to clearly identify and define the health care pod technological system. Accordingly, the first goal will be:

(1)  to define a common taxonomy for these systems and their different typologies, together with a desk analysis of available solutions worldwide.

In relation to the existing systems, one of the most recent health care pod solutions was selected and tested in a real-world setting to verify from the general population the demand for the awareness of personal health status through a set of quantitative measurements related to the main factors. Therefore, the second and third goals will be:

(2)  to show preliminary data relevant to the introduction of a health care pod solution in a specific scenario of large malls as a distribution point for self-opportunistic screening;
(3)  to discuss the application and potential impact of such technology in preventive medicine, according to these personal and collective pictures of individual or societal health status.

## 2. Materials and Methods

### 2.1. Taxonomy and Definition

The continuous demand for new health services led to the recent development and deployment into the market of integrated solutions to enable the user to have a fast health check with respect to several human health domains, thanks to a set of easy-of-use measuring devices integrated into a standalone station. This concept has been previously called health pod. A similar acronym was recently used to identify points of dispensing (POD) that are community locations at which state and local agencies dispense and administer medical countermeasures to the public, such as vaccines or others [18]. However, this definition does not fit with the systems discussed here.

For preventive medicine, health pods use 'non-invasive' methods for monitoring several physiologic parameters through an assisted or self-based and interactive (i.e., user interface driven) procedure to guide the user and avoid errors. In some cases, the output is represented by a report, including the measured parameters, their interpretation based on well-being and lifestyle, and a set of indicators related to associated risks for the most relevant pathologies. Other services (e.g., video or teleconsultation) can also be implemented through these systems [21].

The recent introduction of these "health pods" requires their precise definition and classification. Up to now, in the scientific literature, no specific definition of such systems

has been provided. In general, a pod is an encapsulated space in the form of a capsule. In relation to health processes, we can identify two basic scenarios: with or without the presence of a healthcare operator, both with the option of having available biomedical technologies for any kind of measurement of physiological parameters or not. This technological point of health could be enclosed with some kind of walls or separators to create a small, enclosed space; in this case, the concept of "health kiosks" has been introduced. Conversely, in the case of a system without walls, in an open environment, and in a vertical configuration, the health pod has the shape of a totem.

From all the above, in the frame of e-health services, the following definition could be identified and proposed:

A health pod is a small space, set in public or private environments, equipped with some medical devices where people can carry out check-ups, screenings, or follow-ups and eventually receive remote visits or clinical consultancy.

Included in this definition, we have the two previously mentioned typologies of health pod: (a) health kiosk, when walls or separators divide the space in an enclosed private one where, through the presence of biomedical technologies, some measurements or procedures are carried out (Figure 1) with or without the presence of a healthcare professional; (b) health totem, usually in a vertical configuration or placed onto a desk, again with some measuring devices for some health-related parameters or signal (Figure 2).

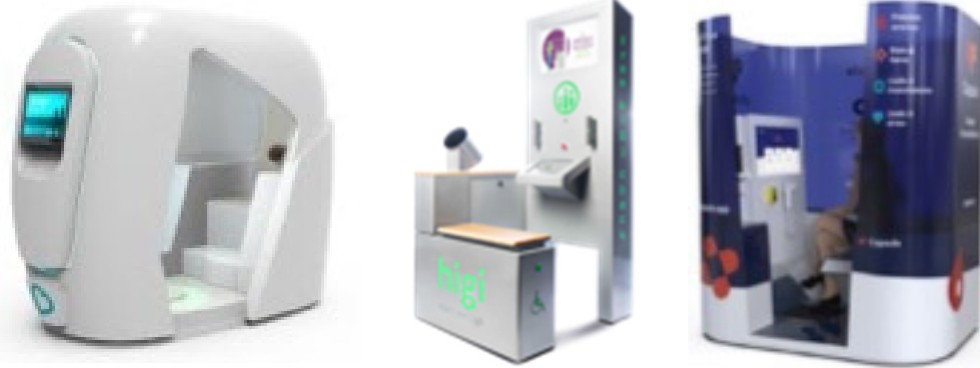

**Figure 1.** Example of kiosks: Bodyo (**left**), Higi (**center**) and CAPSULA (**right**).

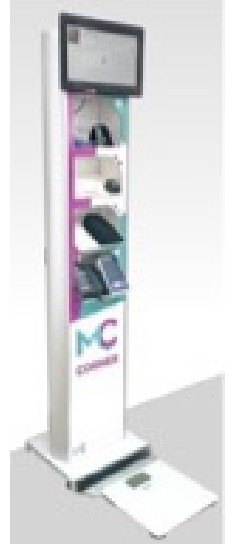 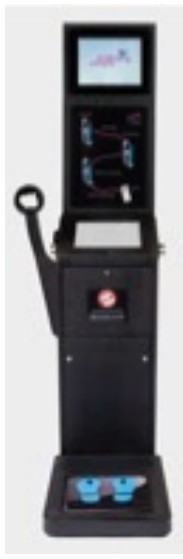 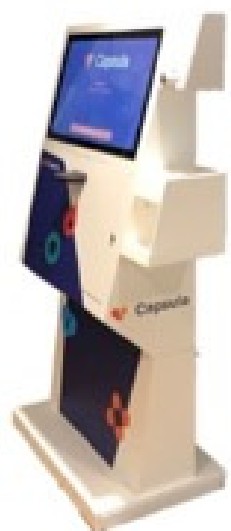

**Figure 2.** Example of totem solutions.

As said, the last variable is the presence (or not) of a healthcare operator to support or lead the procedure. In case of absence, the subject has to carry out the procedure in

self-mode. In this case, a very efficient, intuitive, and engaging user experience/user interface (UX/UI) is needed to avoid procedural errors and obtain reliable measurements.

*2.2. Desk Analysis*

A deep market survey through a desk analysis was conducted focusing on these systems in the clinical field and in general web using Pubmed and Google engines. The following keywords were used: "health" + "pod" + "kiosk" + "totem." Display-only solutions were excluded, while only systems with measuring devices and/or telehealth services were included. Similar research was also conducted on the patent portal Espacenet.

Three main categories of solutions were identified: (1) systems providing health screening in self-mode, (2) systems providing health screening in assisted mode, and (3) systems or kiosks providing health services in remote mode. The analysis also considered the implementation of measurement of signals and the eventual certification as a medical device.

**3. Results**

*3.1. Market Analysis of Health Pod Solutions*

In Table 1, the global analysis of current health pod solutions is provided.

Most of the existing solutions are bulky in terms of procedures, assistance needed for clinical-grade measures, modularity, and technological integration, and there are limitations to offered services; for these reasons, their adoption is still minimal despite a market with huge potential.

A representative system that is the most exploited solution is represented by HIGI [22], with about 11,000 kiosks installed in the USA in pharmacies and other shops, by which the customer can perform two tests: blood pressure (BP) and weight/body fat. The user has to register in the platform, and the business model is mainly in selling big data; in this way, more than 300 M tests were performed, thus providing an idea of customer needs and demand.

A new system is trying to overcome the previously listed limitations: CAPSULA is the last one in terms of time on the market, with the specific characteristic to be certified as a medical device in Europe in Class II and to allow self-measurements of a quite large set of biomedical parameters (BP, ECG 1-lead with heart rate (HR) and HR variability (HRV) analysis, breathing rate, temperature, pulse oximetry, advanced glycation end products, and weight and body composition through body impedance analysis (BIA)) [23]. CAPSULA has also introduced an interesting new concept: a phygital health pod, where "phygital" means that the user is living a digital experience through a physical interface.

*3.2. Biomedical Measurements and Services Available through Health Pods*

The new generation of health pods is capable of automatically, quickly, and reliably measuring a series of parameters directly associated with several risk factors and diseases, such as:

- blood pressure → cardiovascular diseases;
- heart rate (HR) → cardiovascular diseases;
- HR variability (HRV) → cardiovascular diseases and psycho-physical well-being;
- oxygen saturation (SpO2) → chronic respiratory and cardiovascular diseases;
- respiratory rate (RR) → chronic respiratory and cardiovascular diseases;
- advanced glycation end products (AGEs) → diabetes mellitus and metabolic situation;
- body weight (BW), body mass index (BMI), fat-free mass (FFM), fat mass (FM), total body water (TBW) → all-cause mortality risk from BMI.

In addition to these, other measurable parameters can be added which, although not directly indicative of the most lethal morbid conditions, are nevertheless important for the overall evaluation of health (i.e., strength, hearing, body temperature).

**Table 1.** Analysis of competitors in health pods. In yellow the most interesting initiatives adopting a self-assessment approach are highlighted.

| System | Company | Country | Medical Certification | BP | Weight | Height | BIA | Temperature | Eyesight | Single-lead ECG | SpO2 | COVID Triage | HRV | PWV | BP | Weight | Height | BIA | Temperature | Eyesight | Single-Lead ECG | ECG 12 | SpO2 | Dermatoscope | Otoscope | Hearing | Stethoscope | Blodd Analysis | DNA | Dispensary of medicine | Densitometry | Holter | HRV | Thermal camera |
|---|---|---|---|---|---|---|---|---|---|---|---|---|---|---|---|---|---|---|---|---|---|---|---|---|---|---|---|---|---|---|---|---|---|---|
| | | | General Information | | | | | Self Parameters | | | | | | | | | | | | | Teleconsultation Parameters | | | | | | | | | | | | | |
| **Consult Station** | H4D sarl | France | CE Med IIA-FDA | | | | | | | | | | | | X | X | X | | X | X | X | | X | X | X | X | X | | | | | | | |
| **HIGI** | Higi SH llc | USA | FDA | X | X | | X | | | | | | | | | | | | | | | | | | | | | | | | | | | | |
| **Pursuant Health** | Pursuant Health | USA | FDA | X | X | | | | X | | | | | | | | | | | | | | | | | | | | | | | | | | |
| **CAPSULA** | CAPSULA | Italy | CE Med IIA | X | X | | X | X | | X | X | X | X | | X | X | | X | X | | X | X | X | X | X | | | X | | | | | | X |
| **Health Point** | Health Point SpA | Italy | | | | | | | | | | | | | X | X | X | | X | X | X | X | | | | | | X | X | | | X | X | |
| **Bodyo** | Bodyo | France | ND | X | X | | X | | | X | | | | X | | | | | | | | | | | | | | | | | | | | | |
| **One Minute Clinic** | Ping An | China | | | | | | | | | | | | | X | | | | X | | | | | | | | | | | | | X | | | |
| **Lifestyle check point** | Lifestyle check point | UK | | X | X | X | X | X | | X | | | | | | | | | | | | | | | | | | | | | | | | | |
| **SISU** | Sisu Health group | Australia | | X | X | X | X | | | | | | | | | | | | | | | | | | | | | | | | | | | | |
| **Spotcheck** | Spotcheck | Dubai | FDA-CE | X | X | | X | | | | | | X | X | | | | | | | | | | | | | | | | | | | | | |
| **Tessan** | Tessan | France | | | | | | | | | | | | | X | | | | X | | | | | X | X | X | | X | | | | | | | |
| **Vitalis** | 24aLife | USA, Slovenia | | X | | | X | | | X | X | | X | X | | | | | | | | | | | | | | | | | | | | | |
| **Onmed** | Onmed | USA | | | | | | | | | | | | | X | X | | | | | | | | | | | X | | | | | | | X |
| **MS Fit** | Medicalsoft | Russia | | | X | | X | | | | X | | X | X | | | | | | | | | | | | | | | | | | | | |
| **Health ATM** | Yolo Health | India | | | | | | | | | | | | | X | X | | X | X | | X | X | X | | | X | X | | | | | | | |
| **Wellbeing people** | Wellbeing people | USA | | X | X | X | X | | | | | | | | | | | | | | | | | | | | | | | | | | | |

To complete the general estimate of cardiovascular and metabolic risk, in the future, it would be useful to also measure some parameters from the capillary blood through validated self-analysis systems (total cholesterol, HDL, triglycerides with the calculation of LDL according to the Friedewald formula [24], glycated hemoglobin).

The systematic and transversal use of these health pods in the general and asymptomatic population could help identify many people with one or more risk factors to be able to intercept and manage them in advance. From this perspective, they have the capability to:

- empower patients' self-management of lifestyle and health, thus implementing the first paradigm of prevention by awareness and reducing risk factors (through the personalized coaching delivered through advanced cloud services);
- provide personalized prevention programs, promoting health environments, encouraging physical activity and healthy nutrition through a gamified approach; redefine the patient/caregiver relationship, in which care teams and subjects decide together when to visit live or virtually, reducing time and costs of the traditional healthcare delivery approach, providing immediate and geographically distributed tools for monitoring and evaluating the patients progresses;
- empower patients' self-management of disease, helping patients to increase the level of adherence to their clinical programs and to make the healthy choice the easiest one, at the right time.

### 3.3. Test of Health Pod Application in a Real-World Setting

To verify these assumptions, a pilot test was conducted in collaboration with CAPSULA by placing a single-unit health pod in two malls in the Milan (Italy) surroundings, characterized by a large number of people accessing the structure (the first site (S1) with 20,000+ visitors per day, and the second site (S2) with 5000+ footfalls per day).

The health pod was equipped with four available tests:

- measurement of BP through an automated system;
- measurement of AGEs through spectroscopy in fluorescent light;
- measurement of ECG 1-lead and assessment of stress level through HRV analysis;
- measurement of body weight and composition through BIA.

In this pilot, the subject was able to freely decide in complete autonomy which test to carry out (or all tests) in his/her preferred order.

During 57 days of observation (2 June–28 July 2021) at the two sites, impressive outcomes were registered (Figures 3 and 4); an average of 116 tests per day at S1 and 93 tests per day at S2 were performed in the period, where the number of tests per user was 1.5 at S1 and 2.0 at S2. Participants were equally gender distributed: 51% females and 49% males, thus showing the same demand for prevention among genders in the general population. The participation rate, defined as the number of health pod users on total visitors, was 0.2% during weekdays in S1; this value raised to 0.6% on the weekends. For S2, these rates were respectively 0.4% on weekdays and 0.8% on weekends.

About completion rate, computed as the % of completed tests with respect to the total number of tests initiated, the values are 83.60% for S1 and 90.86% for S2, thus showing very good system reliability and user's commitment due to the engagement strategy through the user interface.

A very interesting datum is related to the redemption rate: CAPSULA offers the possibility to obtain a more detailed description of the results by connecting through the QR code link to the data webpage, where a scientific description of the measured parameters and their normality ranges are indicated. For S1, 43% of the subjects accessed this link, while for S2, this percentage was 25%. These values are really meaningful and represent the quantification of the request for prevention in the general population.

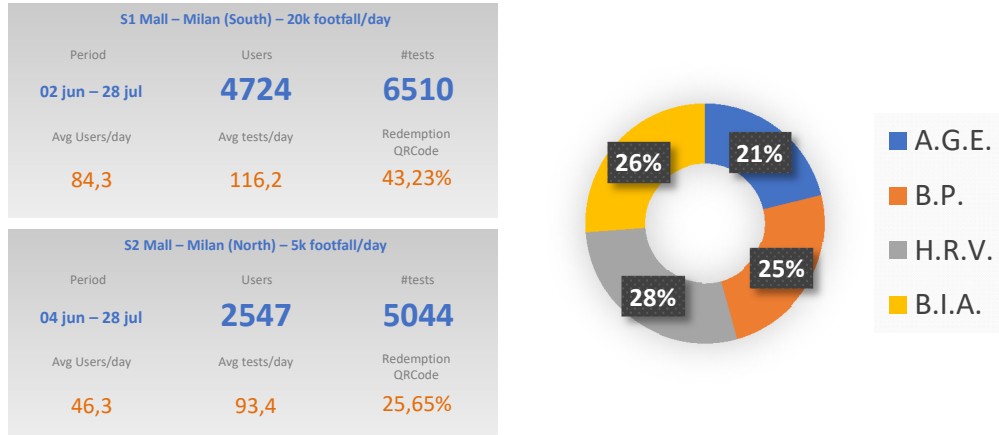

**Figure 3.** Statistics of health pod pilot tests in malls with 20,000+ visitors per day (S1) and 5000+ visitors per day (S2) and total test distribution at the two sites (**right**): A.G.E., advanced glycation end products; B.P., blood pressure; H.R.V., heart rate variability; B.I.A., body impedance analysis.

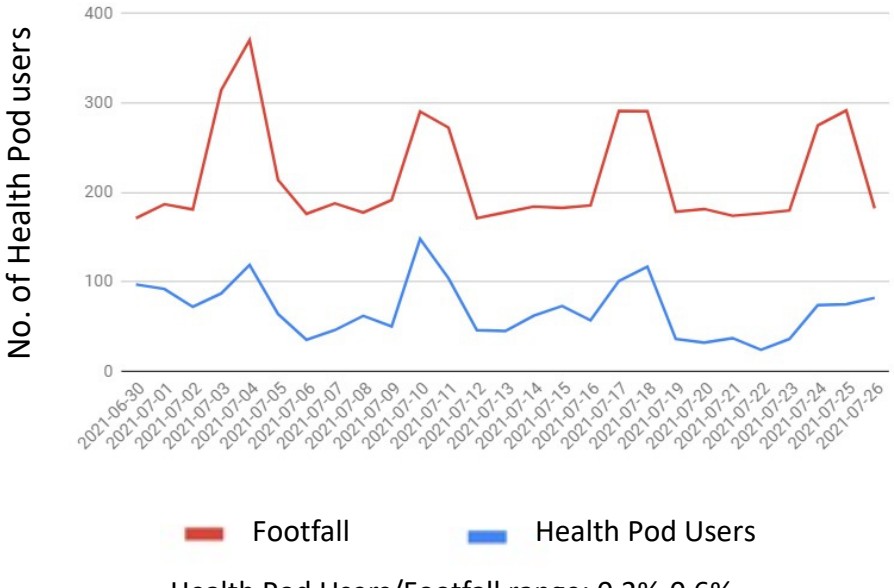

**Figure 4.** Health pod recruitment in tests in respect to total footfall at S1.

The average age (min, max) of the reached population (men: 49 yrs (9, 87); women: 45 yrs (9, 87)) is in line with the expected value that is indicated as the target of people potentially interested in prevention activities (men > 40 yrs, women > 50 yrs [25,26]). It is worth noting that even >65-yrs subjects successfully took part in the tests; this opens the possibility to also carry out through health pod technology mass screening campaigns for specific pathologies, such as atrial fibrillation, which is frequent and asymptomatic in the early phases in these subjects, or hypertension.

The check-up tests carried out by the participants were well balanced in distribution among the different available options. In terms of duration of the examination, for a subject completing the full set of check-ups, an average time of 7 min 15 s was measured, as a result of the average duration recorded for each test (Table 2). This limited-time requirement encouraged the completion of more than a single test in most of the participants.

**Table 2.** Analysis of time for test completion in each domain for site S1 and S2.

| *Check Duration per Typology and site* | Duration Avg (mm:ss) | Duration Max (mm:ss) | Duration Min (mm:ss) |
|---|---|---|---|
| Age Check @ S1 | 00:59 | 05:26 | 00:34 |
| Body Composition Check @ S1 | 02:01 | 09:50 | 01:06 |
| HRV Check @ S1 | 02:45 | 15:40 | 01:49 |
| Vascular Check @ S1 | 01:31 | 05:48 | 00:57 |
| Age Check @ S2 | 00:57 | 03:07 | 00:29 |
| Body Composition Check @ S2 | 02:01 | 05:46 | 01:09 |
| HRV Check @ S2 | 02:40 | 12:33 | 01:52 |
| Vascular Check @ S2 | 01:29 | 05:34 | 00:55 |

These results show an apparent difference between the two sites, specifically in the numbers of users and in the number of performed tests. This is coherent and, in a certain sense, expected because it is correlated with the average number of visitors to each site. The fact that the number of tests was not linearly increased in S1 compared with S2 is due to the time saturation of the health pod. Table 2 shows that the time needed for each test was very similar in both sites, thus demonstrating the good usability and UX of the system. Indeed, with an average of 116.2 tests per day at S1 and an average time of 7 m 16 s, it means that the health pod was continuously utilized by a user approximately for 14 h 5 m, thus covering all the opening hours (8–22, i.e., 14 h). Almost the same situation was present at S2, with a total usage of the health pod of 11 h 6 m. In this location, some periods of non-utilization were registered, in particular in the early morning and during lunch/dinner time, as influenced by the minor number of visitors.

Finally, a global picture of the health status of the tested population is presented (Figure 5), with good general outcomes in all the four investigated areas. BP is well positioned, being in the normality range according to European Society of Hypertension recommendations; the same was observed for AGE, which evaluates metabolic status and pre-diabetes risk.

On the contrary, the BMI demonstrates a certain tendency to be overweight, which is in line with the statistics at the national level. In addition, muscle score, % fat mass, and hydration are coherent with this tendency, which could be explained by low physical activity level, as assessed by % tone, % flexibility, and % dynamism. Finally, the stress level resulting from HRV analysis could be considered slightly elevated.

A more detailed view is presented in the following Table 3 where average results divided by different age ranges are shown. It is possible to observe how the most numerous age range that participated was for both genders between 40 and 54 years old. Again, the health status of the observed population appears good on average. The metabolic assessment reveals a generally coherent AGE score in relation to the age range of the subjects, thus demonstrating to be a good descriptor of the metabolic function, with better (lower) values in females than males. A tendency of an increase in systolic BP was present with age, in particular in females, but still <140 mmHg, as is recommended as the target value for subjects >65 years old [27]. As regards body composition, a small overweight is present in both males and females.

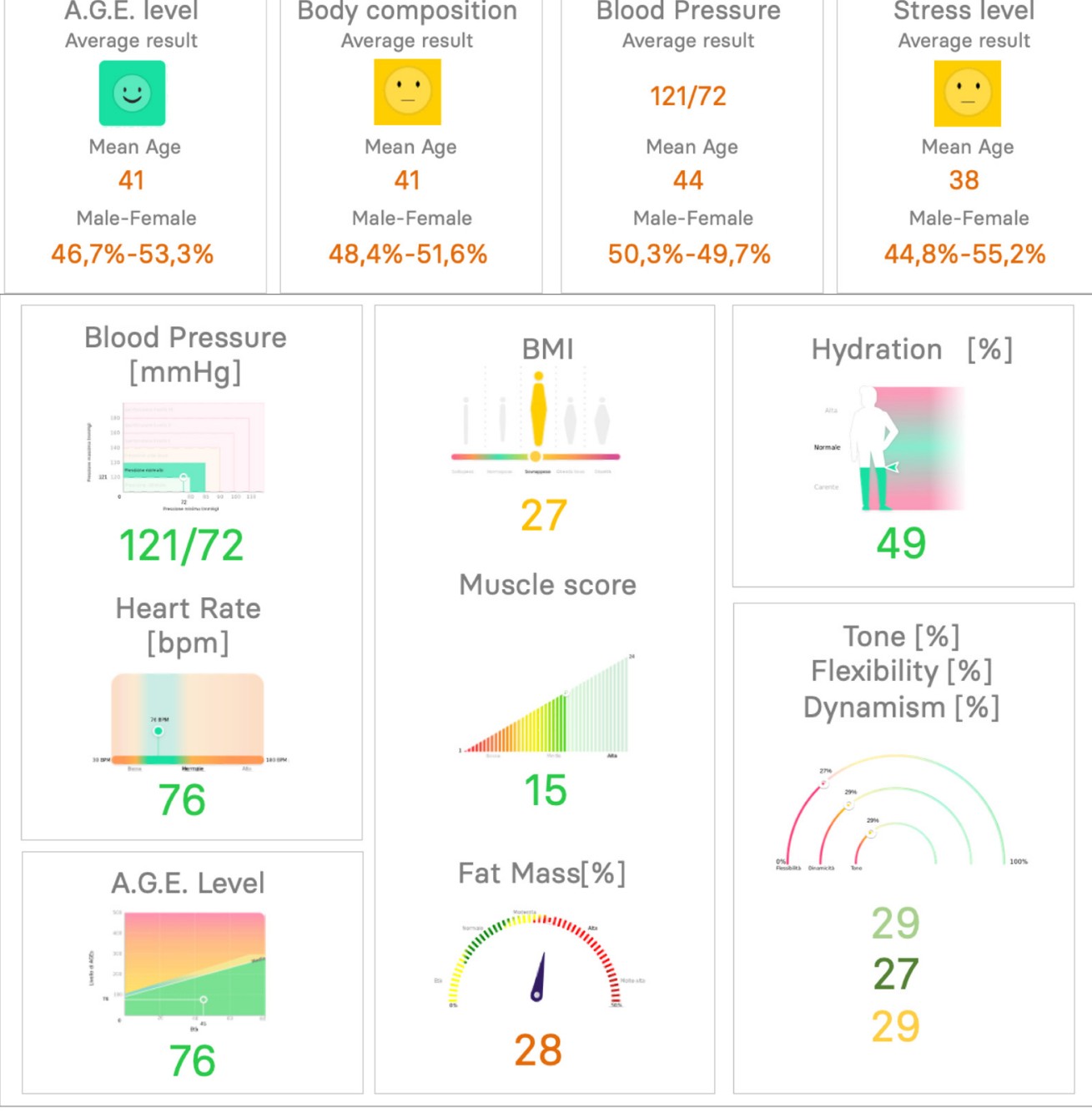

**Figure 5.** Global outcome of the health status of the population that participated in health pod pilot tests at S1 and S2. Semaphoric color-coding of the represented numbers shows the potential difference from the considered normality levels.

Table 3. Health data of male and female subjects divided per age range for the S1 + S2 population.

| Men | | Age | | Blood Pressure | | | | | | Body Composition | | | Heart Rate Variability | | | |
|---|---|---|---|---|---|---|---|---|---|---|---|---|---|---|---|---|
| Age Range | Total No. | Age Avg Value | Sample Size | BP Avg Sys | BP Avg Dia | Avg HR | Sample Size | BMI | % Fat Mass | % Hydration | % Muscle Mass | Sample Size | HRV Tone | % HRV Dynam | % Flexibility | Sample Size |
| 65+ | 758 | 234.3 | 158 | 125.7 | 67.2 | 69.4 | 313 | 27.2 | 25.2% | 54.9% | 71.1% | 158 | 52.4% | 29.6% | 16.5% | 127 |
| 55–64 | 922 | 202.4 | 214 | 126.0 | 77.4 | 74.6 | 253 | 28.1 | 24.7% | 54.4% | 71.5% | 229 | 37.5% | 28.6% | 21.8% | 224 |
| 40–54 | 1814 | 175.2 | 410 | 123.9 | 76.7 | 74.0 | 500 | 27.8 | 23.8% | 54.8% | 72.4% | 455 | 35.8% | 32.4% | 27.4% | 442 |
| 25–39 | 1417 | 143.9 | 318 | 119.9 | 70.2 | 74.8 | 329 | 26.7 | 22.4% | 55.8% | 73.7% | 376 | 34.2% | 41.6% | 40.1% | 391 |
| 18–24 | 875 | 138.6 | 165 | 107.3 | 63.4 | 78.7 | 177 | 23.2 | 28.3% | 49.5% | 68.0% | 263 | 19.6% | 33.6% | 34.6% | 267 |
| TOTAL | 5786 | TOTAL | 1265 | | | TOTAL | 1572 | | | | TOTAL | 1481 | | | TOTAL | 1451 |
| Females | | Age | | Blood Pressure | | | | | | Body Composition | | | Heart Rate Variability | | | |
| Age Range | Total No. | Age Avg Value | Sample Size | BP Avg Sys | BP Avg Dia | Avg HR | Sample Size | BMI | % Fat Mass | % Hydration | % Muscle Mass | Sample Size | HRV Tone | % HRV Dynam | % Flexibility | Sample Size |
| 65+ | 548 | 199.2 | 128 | 128.6 | 65.5 | 72.4 | 179 | 25.3 | 29.4% | 48.1% | 67.0% | 117 | 47.6% | 33.5% | 17.9% | 123 |
| 55–64 | 917 | 189.2 | 215 | 120.1 | 69.1 | 74.9 | 258 | 25.8 | 30.4% | 47.4% | 66.0% | 208 | 36.0% | 30.9% | 23.8% | 233 |
| 40–54 | 1912 | 176.4 | 442 | 113.7 | 67.8 | 75.9 | 485 | 25.6 | 30.1% | 48.0% | 66.4% | 467 | 32.5% | 35.1% | 28.5% | 508 |
| 25–39 | 1593 | 151.1 | 351 | 110.3 | 65.6 | 78.2 | 361 | 24.7 | 29.1% | 48.9% | 67.3% | 423 | 24.5% | 38.6% | 38.4% | 456 |
| 18–24 | 875 | 138.6 | 165 | 107.3 | 63.4 | 78.7 | 177 | 23.2 | 28.3% | 49.5% | 68.0% | 263 | 19.6% | 33.6% | 34.6% | 267 |
| TOTAL | 5786 | TOTAL | 1301 | | | TOTAL | 1460 | | | | TOTAL | 1478 | | | TOTAL | 1587 |

## 4. Discussion

Health pods are a new concept that aim at gaining an ambitious positioning in the future of healthcare services. These systems are spreading as solutions for distributed health assessment points, useful for population screening or for chronicity management, and even in the COVID era, as a sensing point to assess and monitor the epidemiologic diffusion of the pandemic through specific parameters. The main idea is to provide in easily accessible sites quantified information about the health status of users and to engage them in improving and taking care of their health, thus implementing in the real-world the concept of distributed medicine with a clinical service closer to citizens and to where they live.

Hospitals, pharmacies, shopping malls, gyms, and even supermarkets could be furnished with medical pods that can provide the first screening for health problems in a few minutes. It is necessary to point out that these systems cannot provide a diagnosis (which is a medical responsibility of a clinical doctor) but only a set of clinical-grade measurements and their positioning in the frame of a set of normality ranges. As most of the solutions also exploit cloud services, once the user has carried out the measurements, the results are stored in a data cloud that can then be easily shared with healthcare professionals. When the collected information is shared with medical facilities, health pods could enable early diagnosis and prevention services.

From this perspective, health pods could represent a strategic opportunity to build a phygital health ecosystem that empowers people to healthy behaviors and enables the 5P medicine era. In fact, they implement four out of the five dimensions of the P5 medicine vision: participative, preventive predictive, and personalized health management, even thanks to pluri-experts' opinions that will prevent pathologies and predict/reduce risks. A joint marketplace could complete the offer of preventive or consultation or monitoring intervention services, for example, by integrating the health pod system with wearables or AI to have a continuous and real-time lifestyle check and support. The information and the awareness about own health status and the education toward the strategies, actions, and measures on how to improve it represents the main goal of subject empowerment toward a 5P medicine. Indeed, empowerment of people for becoming co-producers and co-managers of their own health starts with awareness, and only through this strategic action can the motivation to prevention could be engaged. This is an active countermeasure to aging decline, chronic disease management, well-being promotion, prevention, and early diagnosis. In fact, the possibility to self-measure parameters related to the presence/absence of pathologies and/or comorbidities, or only as simple risk factors, in relation to the different age ranges, could help identify personalized situations and interventions. According to the lifestyle medicine paradigm, a direct correlation between the signals measured through the health pods and different risk of pathologies (Figure 6 shows the main ones) is known, and their individual assessment could allow designing specific and personalized protocols for prevention strategies. They could even work jointly with the recent concept of personal digital health hub, an integrator and interpreter of useful health information capable of collecting data from different health services and other personalized digital data sources [28] to support and empower patients to take greater control of their health goals in innovative models of care. Moreover, in a prevention vision, this approach requires a redesign of digital health solutions that will allow the co-participation and delivery of high-quality and patient-centered health services in her/his community setting. In this scenario, the health pod could result in an important tool to empower people toward taking care of their own health jointly with the national welfare system and its main actors, in particular in a post-pandemic scenario. Similarly, the possibility to obtain a global picture of the health of the general population for a certain territory/location/group of users could help the national or territorial welfare system to design and plan the most proper strategies to implement efficient healthcare to improve public health.

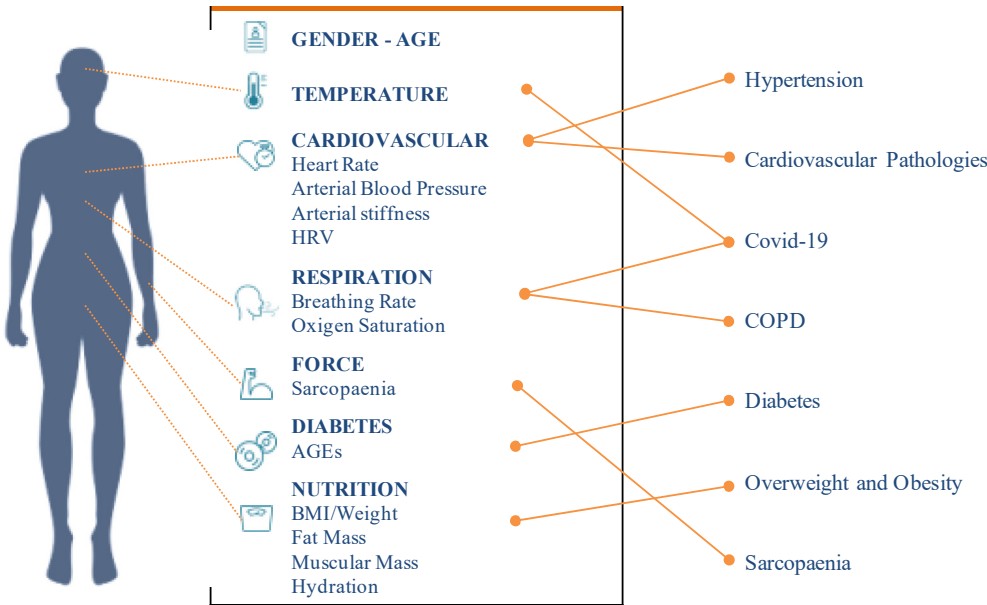

**Figure 6.** Personalized protocols for prevention.

Moreover, in the frame of corporate welfare actions, companies can place the pods in their sites/offices to encourage employees to use them by means of incentives or discounts on selected health-related products or services when certain goals are met.

The European Union recently released "The Economy of Wellbeing" plan as one of the most important priorities for EU in the next years [29], where people's well-being is a principal aim of the European Union. This document specifically highlights the need for putting people at the center of policy with a growth model that should be equitable and sustainable from the outset [30]. Citizens' well-being should drive economic prosperity, stability, and resilience, and vice-versa, in a virtuous loop. This policy orientation and governance approach leverages innovative technologies such as health pods that could allow access for all to health services, long-term care, health promotion, and disease prevention provided by a sustainable health system.

In conclusion, there is a need for a co-creation of digital health models implementing a continuous pathway of care, and we strongly believe that health pods could play a strategic role in this field by filling the existing gap, thus promoting a continuous health path in which each subject could contribute both at a personal and a collective level. Health pods could act as distributed health monitoring stations, thus implementing a diffused point-of-care model that allows the citizen to obtain easy access to available biomedical technologies close to home. In this way, they can become active contributors to their health by keeping monitored their functional indexes in the main health risk domains, thus receiving dedicated and personalized services, including education and even more complex interventions. A properly developed UX design should support overcoming differences and inequalities in digital and health literacy. The COVID pandemic and its long-term COVID or post-COVID implications would increase the demand for similar approaches, in which the measured parameters could be adapted to the needed monitoring based on current technologies.

**Author Contributions:** G.A. produced the original draft and carried out the taxonomy definition, market survey, and data presentation; E.G.C. contributed to the paper aims' definition and data discussion; N.C. provided the clinical overview. All authors provided critical review, feedback, additions, further references, and guidance. All authors have read and agreed to the published version of the manuscript.

**Funding:** This research received no external funding.

**Conflicts of Interest:** Giuseppe Andreoni received consulting fees for the design of the system and scientific advice for technological selection in measurements by CAPSULA srl.

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
