# Peer review of "Digital Health Services through Patient Empowerment: Classification, Current State and Preliminary Impact Assessment by Health Pod Systems"

_applsci, doi:10.3390/app12010359_

Round 1

Reviewer 1 Report

This paper aims at defining their taxonomy, conducts a market and typologies survey, and discusses their potential impact in preventive medicine presenting data of a pilot test carried out placing two health pods in superstore environment to validate the demand and the participation of people to prevention campaign.

 Article interesting! I read the manuscript. It is appreciated that the manuscript is easy to follow and not too long. The message is clear and of interest to the community. I would like to accept the manuscript in the present form.

Author Response

We thank you the reviewer for the appreciation to our work and related publication.

Reviewer 2 Report

Thank you for the opportunity to review this manuscript.

Although it is an interesting work undertaken by the authors of this manuscript, however I have few comments, hopefully some of those will be constructive and further improve the quality of the manuscript.

Overall comment- The current flow of the manuscript requires bit of working as it is difficult to follow and had to read couple of time to fully comprehend concepts which can be simplified such as divide the introduction and results sections into subsections. So that it can flow in a logical manner tackling one concept at a time.

Abstract- key findings from the results sections do not apparently aligns and there is a scope to further strengthen

Introduction- as mentioned above, please divide this section into subsections and tackle different ideas and link it appropriately so that it is easier for the readers to follow. As of now this seems to be putting together different ideas and there is an assumption that they are linked and the readers with different orientation will be able to comprehend.

"The aim of this paper is threefold"- please clarify if these are specific objectives whereas overall aim could be different.

Materials and methods- Please align the methods with the stated aims/objectives and give more detail about the second aim/objective. How the preliminary data was made available to the authors? Please clarify if the authors or some of them were part of the primary study? Bit of elaboration will help other researchers to design similar kind of study.

Results- The structure of this section can be improvised by presenting the findings in subsections and align with objective of this paper which could improve the readability of the manuscript.

Discussion- This can be strengthened by talking more about defining the contours of empowerment, what is the role of education and behavior change? Please discuss more about facilitating education, role of health literacy, eliteracy, health inequalities in relation to access to digital technologies. Please elaborate on limitations of these health pod systems and how they can be improvised further based on advancing knowledge post COVID-19 pandemic.

You can refer to couple of recent publications about how digital health solutions needs to take account of behavior change education to offer an integrated and holistic care for multimorbid conditions as you mentioned about increasing burden of NCDs and ageing population and the role of patients, their carers (formal and informal as networked units as part of the ecosystem)- all needs to be empowered and treated as part of health workforce. Therefore, there is a need for a co-creation of digital health model of care rather than one solution which is disjointed with the continuing pathway of care.

  1. Chehade MJ, Yadav L, Jayatilaka A, Gill TK, Palmer E. Personal digital health hubs for multiple conditions. Bull World Health Organ. 2020 Aug 1;98(8):569-575. doi: 10.2471/BLT.19.249136. Epub 2020 Jun 2. PMID: 32773902; PMCID: PMC7411320.
  2. Yadav L, Gill TK, Taylor A, De Young J, Chehade MJ. Identifying Opportunities, and Motivation to Enhance Capabilities, Influencing the Development of a Personalized Digital Health Hub Model of Care for Hip Fractures: Mixed Methods Exploratory Study. J Med Internet Res. 2021 Oct 28;23(10):e26886. doi: 10.2196/26886. PMID: 34709183.
  3. Yadav L, Gill TK, Taylor A, deYoung J, Visvanathan R, Chehade MJ. "Context, content, and system" supporting digital health hub (DHH)-enabled models of care (MoCs) for fragility hip fractures: perspectives of diverse multidisciplinary stakeholders in South Australia from qualitative in-depth interviews. Arch Osteoporos. 2021 Nov 6;16(1):167. doi: 10.1007/s11657-021-01031-3. PMID: 34741200.

Reviewer 3 Report

1. This study refers to patient empowerment, but how it is reflected throughout the study process requires further explanation.

2. The depth of the discussion needs to be further strengthened, why there is a  gap between the results of S1 and S2, what are the reasons for this gap, and the reasons behind this phenomenon should be analyzed in depth

3. The population in this study is random, i wonder if limiting the population to the elderly group will draw the same conclusion?
